# Maraviroc Prevents HCC Development by Suppressing Macrophages and the Liver Progenitor Cell Response in a Murine Chronic Liver Disease Model

**DOI:** 10.3390/cancers13194935

**Published:** 2021-09-30

**Authors:** Adam M. Passman, Robyn P. Strauss, Sarah B. McSpadden, Megan Finch-Edmondson, Neil Andrewartha, Ken H. Woo, Luke A. Diepeveen, Weihao Zhao, Joaquín Fernández-Irigoyen, Enrique Santamaría, Laura Medina-Ruiz, Martyna Szpakowska, Andy Chevigné, Hyerin Park, Rodrigo Carlessi, Janina E. E. Tirnitz-Parker, José R. Blanco, Roslyn London, Bernard A. Callus, Caryn L. Elsegood, Murray V. Baker, Alfredo Martínez, George C. T. Yeoh, Laura Ochoa-Callejero

**Affiliations:** 1School of Molecular Sciences, University of Western Australia, Crawley, WA 6009, Australia; a.passman@qmul.ac.uk (A.M.P.); robynpstrauss@gmail.com (R.P.S.); sbm0707@gmail.com (S.B.M.); mfinch-edmondson@cerebralpalsy.org.au (M.F.-E.); neil.andrewartha@murdoch.edu.au (N.A.); kenwoo.gylab@gmail.com (K.H.W.); lukediepeveen@gmail.com (L.A.D.); 21453672@student.uwa.edu.au (W.Z.); 2014rlondon@gmail.com (R.L.); bernard@beexceptional.consulting (B.A.C.); caryn.elsegood@curtin.edu.au (C.L.E.); murray.baker@uwa.edu.au (M.V.B.); george.yeoh@uwa.edu.au (G.C.T.Y.); 2Centre for Medical Research, Harry Perkins Institute of Medical Research, Nedlands, WA 6009, Australia; 21975444@student.uwa.edu.au; 3Proteored-ISCIII, Proteomics Platform, Navarrabiomed, Navarra Health Department, Navarra Institute for Health Research (IdiSNA), Public University of Navarra, 31008 Pamplona, Spain; jfernani@navarra.es (J.F.-I.); enrique.santamaria.martinez@navarra.es (E.S.); 4Institute of Infection, Immunity and Inflammation, Glasgow Biomedical Research Center, University of Glasgow, Glasgow G12 8TA, UK; Laura.Medina-Ruiz@glasgow.ac.uk; 5Immuno-Pharmacology and Interactomics Department of Infection and Immunity, Luxembourg Institute of Health, 4354 Esch-sur-Alzette, Luxembourg; Martyna.Szpakowska@lih.lu (M.S.); Andy.chevigne@lih.lu (A.C.); 6UWA Medical School, University of Western Australia, Crawley, WA 6009, Australia; 7Curtin Medical School, Curtin Health Innovation Research Institute, Curtin University, Bentley, WA 6102, Australia; rodrigo.carlessi@curtin.edu.au (R.C.); N.Tirnitz-Parker@curtin.edu.au (J.E.E.T.-P.); 8Infectious Diseases Area, Hospital San Pedro—Center for Biomedical Research of La Rioja, 26006 Logroño, Spain; jrblanco@riojasalud.es; 9Centre for Comparative Genomics, Murdoch University, Murdoch, WA 6150, Australia; 10Oncology Area, Center for Biomedical Research of La Rioja, 26006 Logroño, Spain; amartinezr@riojasalud.es

**Keywords:** CCL5 chemokine, hepatocellular carcinoma, liver progenitor cells, macrophages, Maraviroc

## Abstract

**Simple Summary:**

Liver stem cells and activated macrophages have been implicated as contributors to liver cancer; hence, reducing their abundance is a potential avenue for therapy. In this article, we demonstrate that Maraviroc, a drug approved for human use, reduces the liver stem cell response and macrophage activation in a mouse model of liver cancer. These findings underline the preventive potential of this drug in liver cancer, a deadly disease for which there are few effective treatments.

**Abstract:**

Maraviroc (MVC), a CCR5 antagonist, reduces liver fibrosis, injury and tumour burden in mice fed a hepatocarcinogenic diet, suggesting it has potential as a cancer therapeutic. We investigated the effect of MVC on liver progenitor cells (LPCs) and macrophages as both have a role in hepatocarcinogenesis. Mice were fed the hepatocarcinogenic choline-deficient, ethionine-supplemented diet (CDE) ± MVC, and immunohistochemistry, RNA and protein expression were used to determine LPC and macrophage abundance, migration and related molecular mechanisms. MVC reduced LPC numbers in CDE mice by 54%, with a smaller reduction seen in macrophages. Transcript and protein abundance of LPC-associated markers correlated with this reduction. The CDE diet activated phosphorylation of AKT and STAT3 and was inhibited by MVC. LPCs did not express *Ccr5* in our model; in contrast, macrophages expressed high levels of this receptor, suggesting the effect of MVC is mediated by targeting macrophages. MVC reduced CD45+ cells and macrophage migration in liver and blocked the CDE-induced transition of liver macrophages from an M1- to M2-tumour-associated macrophage (TAM) phenotype. These findings suggest MVC has potential as a re-purposed therapeutic agent for treating chronic liver diseases where M2-TAM and LPC numbers are increased, and the incidence of HCC is enhanced.

## 1. Introduction

Hepatocellular carcinoma (HCC), a primary malignancy of the liver, is a leading cause of cancer mortality worldwide, and its incidence is increasing in many regions [1]. HCC predominantly arises in the context of chronic inflammatory conditions, most notably, viral hepatitis B (HBV) and C (HCV) [1]. Although infectious agents are the primary cause of liver cancer worldwide, the incidence in Western countries is rising due to the obesity epidemic and non-alcoholic steatohepatitis [2].

Liver tumourigenesis generally arises from continuous parenchymal damage in which hepatocyte cell death drives compensatory proliferation. Within this chronic inflammation context, liver mutations and epigenetic changes accumulate and eventually transform hepatocytes into malignant cells. An understanding of how tissue-intrinsic processes determine the balance of cell proliferation and cell death and how chronic inflammation regulates these processes and causes liver cancer is required to develop effective therapeutics to treat and prevent HCC.

In many settings of chronic liver disease, the regenerative ability of hepatocytes is compromised invoking an alternative path involving liver progenitor cells (LPCs) [3]. LPCs have been identified in HCV infection [4], alcoholic liver disease, HBV infection and genetic hemochromatosis, and importantly, their numbers correlate with disease severity [5]. Furthermore, LPC markers have accurately predicted short-term mortality in patients with alcoholic hepatitis and are correlated with poor prognosis of HCC [6,7]. Collectively, these findings make a strong case for LPCs as a cancer stem cell candidate in hepatocarcinogenesis.

Further evidence of the tumourigenic potential of LPCs has been demonstrated by the tumourigenic transformation of LPCs in vitro by forced expression of oncogenes [8]. MYC over-expression in the liver induces HCC, but when the promoter regulating its expression is switched off, LPC-like cells differentiate into hepatocytes and cholangiocytes [9] suggesting they were the source of the cancer. Moreover, extinguishing p53 expression in an LPC line transforms them into hepatoma cells [10], and tumourigenic LPCs can be isolated from p53 null mice placed on a choline-deficient, ethionine-supplemented (CDE) diet [11]. Collectively these findings suggest that LPCs are a potential source of HCC.

Tumour-associated macrophages (TAMs) are a critical component of the tumour microenvironment (TME). TAMs are macrophages present in close proximity to tumour cells and play important roles in influencing the host immune response to cancer. Macrophages, like many other immune effector cells, exist as multiple subtypes with differing expression patterns, surface markers and secretable factors. Generally, resistance to intracellular pathogens and tumours (Th1-driven responses) are mediated by M1-polarised macrophages, whereas M2-polarised macrophages mediate resistance to parasites, immunoregulation, tissue repair and immuno-tolerance to tumours. When present in high numbers, TAMs are associated with poor survival, the promotion of metastasis, angiogenesis and invasion into nearby tissues and vasculature across many cancer types [12,13,14].

Chemokines and their receptors play crucial roles in the initiation and maintenance of inflammation and fibrosis [15,16], as well as in chronic inflammation that leads to tumourigenesis [15]. CCR5, a member of the G-protein-coupled receptor superfamily, is a cognate receptor for several inflammatory chemokines including CCL3 (macrophage inflammatory protein 1 alpha; MIP 1α), CCL4 (MIP-1β) and CCL5 [16]. CCR5 plays a central role in many events related to liver matrix remodelling as a result of its expression in hepatic stellate cells (HSCs), and there is evidence of interaction between HSCs and LPCs, resulting in LPC proliferation and HSC activation [17]. Accordingly, there are high levels of both CCR5 and its ligand, CCL5, in patients with chronic liver disease and fibrosis [18]. T cells, Kupffer cells [19], HSCs [20] and LPCs [21] all express CCR5 in injured liver. Gene targeting or the use of a potent antagonist for the murine CCR5 receptor results in a significant reduction of liver fibrosis [18]; however, the exact role of CCR5, and which CCR5-expressing cell type is involved in HCC, remains unclear. 

Since CCR5 is a main entry point for the human immunodeficiency virus (HIV), several inhibitors have been developed for its use to treat HIV by reducing viral load; the most common being Maraviroc (MVC) [22]. It has been reported that MVC treatment reduces fibrosis progression in HIV/HCV co-infected patients with CCR5 tropism [23].

Our previous studies showed the importance of inflammation in initiating the LPC response; it is attenuated in IL6 and TNF receptor knockout (KO) mice subjected to a CDE diet, with a commensurate decrease in HCC incidence [24]. The LPC response is also diminished in lymphotoxin beta KO mice [25] and Fn14 KO mice in which tumour necrosis factor-like weak inducer of apoptosis (TWEAK) signalling is attenuated and both inflammation and the LPC response are suppressed [26]. Finally, recruitment of inflammatory monocytes enhances inflammation and promotes initiation of LPC proliferation [27].

Our previous study demonstrated the efficacy of MVC in reducing disease progression and increasing survival rates in a mouse model of CDE-diet-induced HCC [28]. We showed MVC significantly reduced mortality, markers of liver injury, apoptosis, proliferation, expression levels of chemokines, fibrosis and hepatic tumour load [28]. This study sought a mechanistic explanation for the anti-hepatocarcinogenic effect of MVC. We demonstrated that through its anti-inflammatory effect, MVC attenuates the LPC response, coinciding with a reduction in HCC development. We concluded that MVC, an approved, well-tolerated and characterised drug, may be re-purposed as a preventative treatment for HCC.

## 2. Materials and Methods

### 2.1. Animals and Animal Model

A total of 61 WT C57BL/6J male mice were purchased from Charles River (Barcelona, Spain). All animals had unrestricted access to food and water during the study. When the animals were approximately 5 weeks old, they were randomly assigned to one of 4 diet groups: (i) Control, (ii) MVC, (iii) CDE diet and (iv) CDE + MVC. Control group mice were fed a choline-sufficient diet (MP Biomedicals, Illkirch, France, SKU 02960414-CF) and tap water, *n* = 10. Mice in the MVC group received the Control group diet, supplemented with 300 mg/L Maraviroc (MVC, Pfizer, New York, NY, USA) in the drinking water, *n* = 11. For the correct MVC dose in mice, an interspecies allometric scaling factor of 12.3 was used, resulting in a dose equivalent to a human dose of 300 mg/day [29]. The CDE-diet-treated animals received the choline-deficient diet (MP Biomedicals, SKU: 02960210-CF,), and 0.165% ethionine (Sigma, St Louis, MO, USA) supplemented the drinking water, *n* = 20. Finally, the CDE + MVC-treated animals were fed with the same diet as the CDE group but received MVC in the drinking water at the same concentration as the MVC group, *n* = 20. All surviving animals were sacrificed at week 16. Tissue pieces were fixed in buffered formalin (10%) for histological analysis or snap-frozen in liquid nitrogen for biochemical and molecular analyses.

### 2.2. Serum Levels of Alanine Amino Transferase

Blood samples were collected from all surviving animals on week 16 during final sacrifice. Levels of liver damage markers (transaminases) were measured using standard assays in an automatic biochemical analyser (Cobas C711, Roche, Madrid, Spain).

### 2.3. Immunohistochemical Staining

Immunohistological staining was carried out on 5 µm thick formalin-fixed tissue sections. Antigens were retrieved with 40 µg/mL Proteinase K (Dako, North Sydney, Australia) for PanCK and SOX9, 20 µg/mL Proteinase K for F4/80 and EnVision FLEX target retrieval solution (Dako, Cat. GV805) for CD45 immunohistochemistry. Endogenous peroxidases were blocked with 3% H_2_O_2_. Sections were then incubated with DAKO Serum-Free Protein Block (Dako, Cat. X0909), prior to application of PanCK, SOX9 or F4/80 or CD45 (Dako, Cat. Z0622 at 1/400, Merck, Darmstadt, Germany, Cat. AB5535 at 1/500 or Bio-Rad, Gladesville, Australia, Cat. MCA497 at 1/100, respectively) overnight at 4 °C. Other sections were incubated with CD45 monoclonal antibody (BD Biosciences, San Diego, CA, USA, Cat. 550539; 1:10 dilution) overnight at room temperature. PanCK and SOX9 were detected with the LSAB+ kit (Dako, Cat. K5001). F4/80 and CD45 were detected with biotinylated anti-rat IgG (Dako; 1:100 dilution) followed by HRP-conjugated streptavidin (LSAB+ kit). All stainings were visualised with DAB + substrate (Dako, Cat. K0690) as per the manufacturer’s instructions.

### 2.4. Quantitation of Immunohistochemical Staining

For analysis, stained slides were scanned at 40× magnification using the Aperio Scanscope XT instrument (Vista, CA, USA). PanCK+ and SOX9+ were quantitated using the “Positive Pixel Count” algorithm (Aperio ImageScope software) as previously published [30]. Briefly, positively stained pixels were calculated as a percentage of total pixels, yielding % pixel positivity. For analysis of CD45 and F4/80 staining, ImageScope was used to evaluate 20 fields (20× magnification) per sample; these were saved as TIFF files and directly imported into inForm for analysis. Algorithms were created and verified according to parameters and procedures outlined in Appendix A.

### 2.5. LPC Cell Lines and Cell Culture

The PIL2, PIL4 and BMOL LPC cell lines used in this study were isolated from the livers of mice placed on the CDE diet [11,31,32] and cultured in Williams’ E Medium, supplemented as previously described [30]. Supplements were reduced to 2% fetal bovine serum (FBS), 10 ng/mL epidermal growth factor (EGF), 15 ng/mL insulin-like growth factor II (IGF-II) and 0.25 U/mL Humulin R for three consecutive passages prior to use.

### 2.6. Isolation and Culture of Murine Bone Marrow Macrophages (BMMOs)

BMMOs were isolated from C57BL/6J mice by flushing the femur and tibia with PBS. The bone marrow cells were resuspended in RPMI-1640 (Gibco, Paisley, U.K., Cat. 31870) containing 1% L-glutamine, 1% Pen/Strep, MEM, 1X NEAA, 1% pyruvate and 20% FBS and 20 ng/mL CSF-1. Cells were incubated for 10 days at 37 °C and 5% CO_2_ with medium change every 3–4 days.

### 2.7. RNA Extraction and Gene Expression Quantification

Liver, LPC and BMMO samples were homogenised in QIAzol and phase-separated following the addition of chloroform and centrifugation at 12,000× *g* for 15 min. The aqueous (RNA) phase was extracted and added to an equal volume RNeasy Mini kit RLT buffer (QIAGEN, Valencia, CA, USA, Cat. 74104). Total RNA extraction then proceeded according to the RNeasy Mini Kit instructions. Reverse transcription of 5 µg total RNA was performed with Tetro Reverse Transcriptase according to manufacturer’s instructions (Bioline, Eveleigh, Australia). Quantitative RT-PCR was performed using the LightCycler^®^ 480 Probes Master kit (Roche) and primers (Appendix A) with corresponding probes as determined using the Universal ProbeLibrary Assay Design Center. All results were normalised to the expression of the *Taf4a* or *Gapdh* housekeeping gene.

### 2.8. Western Blotting and Protein Quantitation

Prior to harvest, LPCs and BMMOs were serum-starved for 4 h. Afterwards, cells (5 × 10^5^ cells/well) were preincubated with 10 µM MVC for 30 min and/or 50 ng/mL recombinant murine CCL5 (R&D Systems, Lancater, CA, USA) for 15 min. Cellular proteins were extracted, and Western blot analysis was performed. Liver tissue and LPC lines were lysed in DISC Lysis Buffer (150 mM NaCl, 2 nM EDTA, 1% Triton-X, 10% glycerol, 20 mM Tris, pH 7.0) supplemented with 1X Complete Protease Inhibitor Cocktail (Roche Diagnostics, Castle Hill, Australia, Cat. 11697498001), 10 mM sodium fluoride, 2 mM sodium pyrophosphate, 1 mM sodium molybdate and 5 mM-glycerophosphate. Protein content of lysates was quantified using the Bradford Protein Assay (Bio-Rad, Gladesville, Australia, Cat. 500-0006). Lysates (60 µg) were boiled in sample buffer (2% (*w*/*v*) sodium dodecyl sulphate (SDS), 10% (*v*/*v*) glycerol, 62.5 mM Tris pH 6.8, 0.02% (*w*/*v*) bromophenol blue, with 1% (*v*/*v*) 2-mercapto ethanol) and separated by SDS polyacrylamide gel electrophoresis. Electrophoresed proteins were transferred to Hybond C-extra membrane (GE Life Sciences, Buckinghamshire, U.K., Cat. RPN303E) using the wet-transfer method and Criterion blotter (Bio-Rad) at 110 V for 45 min at 4 °C in transfer buffer (25 mM Tris-base, 192 mM glycine, 20% (*v*/*v*) methanol). Once transferred, the membranes were blocked using 5% (*w*/*v*) skim milk powder in Tris -buffered saline-Tween-20 (TBST; 25 mM Tris pH 7.5, 150 mM NaCl, 0.1% (*v*/*v*) Tween-20) before blotting overnight with anti-E-cadherin (Cell Signaling Technology^®^, Danvers, MA Cat. 3195; 1:5000), anti-M2PK (Cell Signaling Technology^®^, Cat. 3198S; 1:1000), anti-CD133 (Abnova, Jhongli City, Taiwan, Cat. PAB12663; 1:1000), anti-GAPDH (Cell Signaling Technology^®^, Danvers, MA, USA, Cat. 5174; 1:18,000), anti-CD68 (Biorad, Hercules, CA, USA Cat. MCA1957T; 1:1000), anti-YM1 (R&D, Cat. AF2446; 0.2 μg/mL), anti-pAKT (Cell Signaling Technology^®^, Cat. 4060; 1:1000), anti-pSTAT3 (Cell Signaling Technology^®^, Cat. 9145; 1:1000) and anti-pERK (Cell Signaling Technology^®^, Cat. 4370; 1:1000) at 4 °C and anti-AKT (Cell Signaling Technology^®^, Cat. 9272; 1:1000), anti-STAT3 (Cell Signaling Technology^®^, Cat. 9139; 1:1000) and anti-ERK (Cell Signaling Technology^®^, Danvers, MA, USA, Cat. 4695; 1:1000) at room temperature. HRP-linked secondary antibodies to rabbit (GE Life Sciences, Westborough, MA, USA, Cat. NA9340), goat (Jackson ImmunoResearch, Suffolk, U.K., Cat. 112-035-003) or mouse (GE Life Sciences, Westborough, MA, USA, Cat. NA9310) were used at a 1:5000 dilution. Whole blot images are shown in Appendix A.

### 2.9. Proliferation Assay

LPCs were plated in 96-well plates at a cellular density of 2000 cells per well in 50 µL of medium, supplemented with 5% fetal bovine serum. The next day, cells were preincubated with 10 µM Sorafenib, 1 µM MVC for 30 min and/or recombinant murine 50 ng/mL CCL5 (R&D Systems) for 15 min. With treatment, the total volume of each well was made up to 200 µL before assessment of growth using the CELLAVISTA instrument (SYNENTECH GmbH). Two cell confluency readings (ideally 12 h apart) were taken per day for at least 4 days or until cell growth plateaued. The confluency percentage for each timepoint for each well was generated by the CELLAVISTA software and imported into an “XY” GraphPad Prism spreadsheet. The exponential growth phase of each well (as defined by data points that best fit an exponential growth equation) was used to determine doubling time (i.e., the time taken for the area occupied by cells to double). These doubling times were generated by GraphPad Prism after fitting the confluency data to an exponential growth equation. Average doubling times and statistics were then obtained by moving the data to a “column” spreadsheet within GraphPad Prism. Lower average doubling times reflect a more rapid proliferation.

### 2.10. RNA-Seq

Total RNA was extracted using the RNeasy Mini Kit (QIAGEN, Hilden, Germany Cat. 74104) and quantitated using Qubit^®^ RNA BR Assay Kit (Thermo Fisher Scientific, Rockford, IL, USA.) according to the manufacturer’s instructions. Sequencing libraries were generated from 1 μg of total RNA using the TruSeq Stranded mRNA HT Sample Prep Kit (Illumina, San Diego, CA, USA) and sequenced to a depth of ~20–30 million reads per sample using 50-cycle single-end reads. Reads were aligned to the mm10 mouse reference genome using Bowtie 2, and fragments per kilobase of transcript per million mapped reads (FPKM) were calculated using cuffnorm.

### 2.11. Single-Nucleus RNA Sequencing (snRNA-Seq)

Single-nucleus RNA sequencing analysis of Control, CDE and thioacetamide (TAA)-treated mice for 21 days was conducted by Carlessi et al. as part of a liver single-nucleus transcriptomics study, yet to be published. Briefly, 40,748 nuclei from the three treatment groups combined were profiled using the 10× Chromium Single Cell 3′ v3 platform. Nuclei were prepared as in [33], loaded onto a 10× chip A and processed on the 10× Chromium controller. Libraries were prepared as per 10× Chromium Single Cell 3′ v3 workflow instructions and sequenced in an Illumina NovaSeq 6000 with S2 flow cells. Reads were mapped using Cell Ranger 2.1.1 with mm10-2.1.0 reference, and downstream quality control, dimensionality reduction, unsupervised clustering and differential expression analyses were conducted on Seurat v3. Ten individual cell type clusters were identified and manually annotated using cell identity marker genes. The expression levels of *Ccr5* were log2 transformed, then reported as mean ± standard error of the mean (SEM) across cell types and treatment groups.

### 2.12. Measurement of Impedance-Based Wound Healing of Confluent BMMO Cultures

Wound healing was determined in BMMOs using the electric cell-substrate impedance sensing (ECIS) system (Applied Biophysics, Troy, NY, USA) as described previously [34]; 1.2 × 10^4^ untreated or treated (10 µM MVC and/or 50 ng/mL CCL5) BMMOs were grown with 10% fetal bovine serum (FBS)-supplemented culture medium on ECIS electrode arrays (8W1E). The impedance fluctuations of cell attachment and spread were continuously monitored. Impedance measurements were collected at 5 min intervals until confluence was achieved. At confluence, wounding of BMMOs was achieved using a 1400 μA signal at 60 kHz for 20 s. Application of this field results in the death of the cells on the electrode and a rapid drop in impedance. Impedance then increases as cells migrate from the perimeter of the electrode inwards to replace killed cells. The slope of the impedance measurement over time is proportional to the speed of cell migration.

### 2.13. Transwell Migration Assay

Cell migration assays were also performed using Transwells^®^. Twelve-well plates with 8 μm pore size inserts (Corning, NY, USA, Cat. 3422) were seeded with 5 × 10^4^ BMMOs in 200 μL of media without FBS ± MVC (10 uM) were added to the upper compartment. Five hundred microliters of DMEM without FBS containing 50 ng/mL CCL5 was added to the lower compartment. BMMOs were incubated in Transwell plates at 37 °C and 5% CO_2_. After 24 h, the insert was taken out, and BMMOs on the lower side of the insert filter were quickly fixed in 5% glutaraldehyde (Sigma, St Louis, MO, USA.) for 10 min then stained with 1% crystal violet (Sigma, St Louis, MO, USA) in 2% ethanol for 20 min. Images of the lower side of the filter were recorded under a microscope (Leica CTR4000, Wetzlar, Germany). The extent of migration was determined by calculating membrane coverage using image thresholding in the ImageJ software (National Institutes of Health, Bethesda, ML, USA).

### 2.14. β-Arrestin Recruitment

Recruitment of β-arrestin-1 to human and mouse CCR5 (hCCR5 and mCCR5) induced by human and mouse CCL5 chemokines (hCCL5 and mCCL5) was monitored by NanoLuc complementation assay (NanoBiT, Promega, Madison, WI, USA), as previously described [35,36,37]. Briefly, 5 × 10^6^ HEK293T cells were seeded in 10 cm culture dishes and, 24 h later, co-transfected with pNBe vectors encoding human or mouse CCR5 C-terminally fused to SmBiT and β-arrestin-1 N-terminally fused to LgBiT. Twenty-four hours post transfection cells were harvested, incubated for 25 min at 37 °C with 200-fold diluted Nano-Glo Live Cell substrate and distributed into white 96-well plates (5 × 10^4^ cells per well). Cells were then treated with MVC for 20 min at room temperature at concentrations ranging from 0.05 nM to 1.11 µM for hCCR5 and from 4.57 nM to 50 µM for mCCR5 then stimulated with human or mouse CCL5 (5 nM). β-arrestin recruitment to hCCR5 and mCCR5 was evaluated by measuring bioluminescence with a Mithras LB940 luminometer (Berthold Technologies, Bad Wildbad, Germany).

### 2.15. Statistical Analyses

Data are presented as means and standard errors of the means (SEM). Statistical analyses were performed using Mann–Whitney U-test unless stated otherwise. For the impedance assay, the growth phases were fitted by simple linear regression with the test for significant differences in slopes selected. All statistics were performed in GraphPad Prism 5 software, San Diego, CA, USA.

## 3. Results

### 3.1. Maraviroc Inhibited Human and Mouse CCR5

Maraviroc (MVC) is a drug designed to block the entry of HIV into human cells, and its interaction with human CCR5 is well established [38], but whether it affects mouse CCR5 has not been formally demonstrated. Binding of ligand to CCR5 results in β-arrestin-1 recruitment to the receptor. We showed that β-arrestin-1 recruitment was inhibited by MVC, demonstrating that it inhibited mouse CCR5. However, inhibition of mouse CCR5 by MVC (IC50 1.9 µM) was 76 times less effective than its inhibition of human CCR5 (IC50 25 nM) (Appendix A).

### 3.2. MVC Treatment Significantly Reduced CDE-Diet-Induced Liver Progenitor Cell Proliferation

It has been reported that LPCs express CCR5 [21]. Therefore we explored the possibility that MVC (by inhibiting CCR5) could reduce LPC expansion and subsequent HCC development. In control and MVC-treated mice, PanCK+ cells were restricted to a single layer lining the bile ducts and constituted 0.20 and 0.14% of total pixels, respectively (Figure 1A,B,E). In contrast, CDE diet administration induced large numbers of non-ductal, PanCK+ LPCs, which constituted 3.3% of the total pixels—a 1553% increase (*p* = 0.0286) (Figure 1C,E). Co-administering MVC with the CDE diet significantly (*p* = 0.0081) reduced the LPC response by 54%, to 1.5% of total pixels (Figure 1D,E). The LPC response of each individual liver, as judged by PanCK levels, positively correlated with the extent of liver damage indicated by serum alanine aminotransferase (ALT) activity (r^2^ = 0.69; Appendix A).

The extent of LPC abundance in the liver was also gauged by quantification of mRNA levels of LPC-associated markers (Figure 2). *Ck19* is commonly used to identify LPCs, and its expression increased by 3759% in the CDE diet compared to controls. MVC treatment reduced *Ck19* expression by 48% when compared with the CDE group (Figure 2A). LPCs possess stem-cell-like properties, and thus the expression of stem-cell-associated markers such as *Cd133* (Figure 2B), *Sox9* (Figure 2C), *Ncam1* (Figure 2D), *Cd24a* (Figure 2E) and *M2pk* (Figure 2F) was also evaluated. These markers were significantly increased by 11,894%, 881%, 6216%, 2544% and 1093%, respectively, in mice subjected to the CDE diet relative to control levels. Expression values were decreased by 71%, 67%, 66%, 82% and 71% when mice on the CDE diet were also treated with MVC. When evaluated by immunohistochemistry (IHC), SOX9 pixel positivity increased by 9621% in the CDE diet relative to the control. Treatment with MVC reduced SOX9 pixel positivity by 53% in the CDE+MVC group relative to the CDE diet group (Appendix A). No significant differences were observed between the livers of the control diet and the control + MVC diet with respect to all markers assessed. The expression of these genes was compared with LPC abundance, as assessed by PanCK staining quantitation, obtaining a positive correlation (r^2^ = 0.58–0.82; Appendix A).

The CD133 and M2PK markers were further assessed by Western blotting to confirm their protein abundance was increased by CDE, and this was attenuated by MVC (Figure 3A). Quantification of the protein bands (Figure 3A) showed that, compared to controls, CDE administration, on average, increased the abundance of these LPC-associated proteins by 1792% and 628%, respectively. Co-administration of MVC reduced expression of these markers by 74% and 53%, respectively, in the CDE+MVC group relative to the CDE diet group (Figure 3B,C). Taken together, the increase in LPC number in response to CDE diet administration was reduced when MVC was co-administered, as determined by immunohistochemistry, LPC-specific mRNA expression and protein abundance.

### 3.3. MVC Reduced CDE-Induced Phosphorylation of AKT and STAT3

Proliferation and migration of activated LPCs are considered key events in hepatic regeneration, and these processes are mediated by phosphorylation of AKT, STAT3 and ERK [39,40]. These pathways are downstream of CCR5 signalling and as such were investigated as a potential mechanism for the MVC-induced LPC attenuation. The amount of phosphoAKT, phosphoSTAT3 and phosphoERK (pAKT, pSTAT3 and ERK) relative to their total protein levels was examined in lysates from each of the four experimental groups (control, MVC, CDE and CDE+MVC). There was a trend toward increased phosphorylated AKT; which was enhanced (283%) in livers from CDE mice relative to controls (*p* = 0.057) and was attenuated (32% reduction) in CDE+MVC-treated mice relative to CDE levels (*p* = 0.028) (Figure 4E,F). Phosphorylated STAT3 was increased (221%) in CDE livers relative to controls (*p* = 0.057) and was diminished (45% reduction) in the CDE+MVC group compared to CDE levels (*p* = 0.028) (Figure 4A,B). No significant change was observed for pERK (Figure 4C,D). These signalling changes were accompanied by global proteostatic derangements in which MVC reverted multiple CDE-induced changes at proteome level, modulating wide-ranging biological functions such as amino acid, fatty acid and carbohydrate metabolism and detoxification (Appendix A). Given these results, pAKT, pSTAT3 and pERK levels were examined in LPCs and BMMOs following treatment with murine recombinant CCL5, in the presence and absence of MVC. Treatments did not significantly alter pAKT and pSTAT3, in either cell type, or pERK levels in BMMOs (Appendix A). Whilst CCL5 treatment increased pERK expression in LPCs, MVC did not abrogate this response, indicating that MVC had little to no effect on these signalling pathways in these cells.

### 3.4. CCL5 and MVC Did Not Alter Proliferation of LPC Lines

To further investigate whether CCL5 and/or MVC directly affect their growth, LPCs were treated with CCL5 and/or MVC, and their proliferation was measured using image-based confluency determination. Under the conditions tested, cell growth rates were unchanged compared to untreated cultures (Appendix A). In contrast, the doubling time of LPCs increased 2.5-fold when treated with Sorafenib as a positive control (*p* = 0.0313; Appendix A).

### 3.5. Ccr5 Was Expressed by CDE Liver and Macrophages but Not by LPCs

Treatment with MVC clearly attenuated the LPC response, but LPCs did not display changes in AKT, STAT3 or ERK signalling in response to MVC (Appendix A). As such, we tested the expression of the MVC receptor, *Ccr5*, in these cells, hepatocytes and liver lysates. By RNA-Seq (Figure 5A), RT-PCR (Figure 5B) and snRNA-Seq (Figure 5C), LPCs expressed no or very low levels of *Ccr5*. Accordingly, it was likely that MVC was working through an intermediary to affect the LPC response. Primary hepatocytes expressed only a small fraction of the total *Ccr5* expressed in liver lysates (Figure 5A), indicating a non-parenchymal cell source was largely responsible for the expression of this receptor, particularly in the CDE diet which is known to be inflammatory. Indeed, we observed a slight 1.3× increase in *Ccr5* expression in the CDE diet relative to normal liver (Appendix A). Macrophages are known to be intricately involved with the LPC response; thus we tested *Ccr5* expression levels in BMMOs. The BMMOs expressed *Ccr5* at far greater levels than LPC cell lines or liver lysates (Figure 5B) indicating that they could be the intermediary resulting in attenuation of the LPC response.

Moreover, by snRNA-Seq we further investigated *Ccr5* expression in liver cell types. *Ccr5* was only detected in reliable amounts in myeloid cells (including different types of macrophages, monocytes and DCs), while T and NK cells also expressed *Ccr5* but in smaller amounts. All other cell types in the liver, such as LPCs, did not show consistent *Ccr5* expression levels (Figure 5C).

### 3.6. MVC Treatment Significantly Reduced the Inflammatory Response to the CDE Diet

As the CDE diet induces an inflammatory response [24], the number of CD45+ inflammatory cells and F4/80+ macrophages was examined in CDE-diet (Figure 6E,F) and CDE+MVC-treated (Figure 6G,H) liver sections. The number of CD45+ cells in the livers of CDE-fed mice was significantly reduced by MVC treatment (*p* = 0.041) (Figure 6I). There was a trend towards reduction in the number of F4/80-positive cells by MVC, although their difference did not reach statistical significance (*p* = 0.20) (Figure 6J).

### 3.7. MVC-Treated Macrophages Exhibited Reduced Migration

Migration of BMMOs using an alternate impedance-based wound healing assay was also inhibited (Figure 7C; *p* < 0.0001). CCL5 showed better wound healing capabilities based on more rapid wound closure (migration, slopes of the curves) in comparison to the MVC-treated wells, demonstrating that MVC was capable of influencing macrophage recruitment.

### 3.8. MVC Modified Macrophage Polarization Induced by CDE

In many tumours, infiltrating macrophages are predominantly the “alternative” M2-like type, providing an immunosuppressive environment that encourages tumour growth [41]. Given the anti-tumour effect of MVC, its effect on polarization was investigated with respect to the TAM markers CD68 and YM-1. All TAMs, including both the M1-like and M2-like subtypes, can be identified and quantified by CD68 immunoreactivity [42], whereas YM-1 marks M2 macrophages exclusively [43]. In comparison with controls, CDE administration trended towards increased CD68 and YM1 abundance by 164% and 64% (*p* = 0.057), respectively, suggesting a diet-induced transition to the M2-like polarity. Expression of these markers was 91% and 98% lower (*p* = 0.028), respectively, in mice that were treated with CDE+MVC when compared with the CDE group (Figure 8B,C). The magnitude of these decreases is much higher than the reduction in macrophage numbers observed earlier (Figure 7). As such, these changes signify that MVC actively reversed the CDE-induced transition from the M1-like to the M2-like macrophage phenotype.

## 4. Discussion

Liver cancer is a significant cause of mortality worldwide, and it will acquire increased importance as lifestyle factors that cause liver disease, such as obesity and alcohol consumption, continue to rise [44,45]. Advances in our knowledge of the hepatocarcinogenic process are hampered by a paucity of animal models that mimic the human condition. This, together with the inability to identify early stages of HCC in humans, has resulted in a poor understanding of the target cell for transformation and the molecular mechanisms that underpin the process.

The murine CDE diet model is a fat-induced chronic liver disease model that displays the hallmarks of liver inflammation, fibrosis and ultimately cancer that is associated with HCC, thus recapitulating the process that develops in humans. The consistent theme of our studies based on this model is that a ductular proliferation response is an early event that generates a population of liver progenitor cells, and their abundance correlates with HCC development. In studies where the LPC response is attenuated by reducing the inflammatory response—especially in transgenic mice with a deficient immune system such as in IL6R, TNFR or FN14 knockout mice—the development of HCC is commensurately reduced [24,26]. In humans, the severity of liver-cancer-causing diseases positively correlates with LPC numbers [5], and LPC markers have accurately predicted short-term mortality in patients with alcoholic hepatitis [6]. From a clinical translation perspective, we have previously demonstrated that the commercially available CCR5 antagonist, MVC, was able to reduce HCC development in the CDE model by 72%. No significant difference was observed for any parameter when comparing animals that received a normal diet in the presence or absence of MVC treatment [28]. This confirms that the drug is fairly safe. The mechanism by which MVC reduces HCC development is not known. This knowledge would underpin strategies to reposition this drug for use in patients with liver pathologies to reduce their progression to HCC.

There is evidence that CCR5 promotes the induction and maintenance of liver inflammation in CDE-diet-treated mice as it facilitates the recruitment of immune cells into the liver [21]. The involvement of the CCR5 ligand, CCL5, in cancer and specifically in HCC has been extensively studied. The majority of studies claim a tumour-promoting role for CCL5 [46]. CCL5 levels were highly correlated with disease progression in advanced HCC [18]. Studies attenuating the activities or expression of CCL5 by MVC [28] and in mouse models of HCC [18,47] indicate that CCL5 signalling is directly responsible for HCC development. Furthermore, overexpression of CCL5 promotes tumour growth and disease progression [48].

The mechanism by which the CDE diet induces liver fibrosis and HCC involves damage to the liver parenchyma with a simultaneous blockade of hepatocyte proliferation [49]. This, in turn, induces the production of a large number of LPCs which remodel the hepatic parenchyma [32]. It has been shown that CCL5 facilitates chemotaxis of liver progenitor (oval) cells [21] probably through binding to CCR1 and CCR3, which may explain their abundance in the liver of rodents that had received the CDE diet. Moreover, oval cells may transform into cancer stem cells and directly contribute to HCC [50]. Thus, as MVC acts as a CCR5 antagonist and drastically decreases tumour incidence, it may modulate the LPC response and prevent development of HCC.

Given the association between HCC development and LPC abundance, we examined the ability of MVC to modulate the LPC response. We showed MVC reduces the LPC response based on quantitation of PanCK- and SOX9-positive cells. Alternative approaches to quantify the LPC response based on gene expression for a range of LPC-specific mRNAs, as well as their respective protein abundance, confirmed the staining results. By these criteria, the CDE diet increased the LPC response by 10- to 120-fold above control diet levels depending on the marker examined. Notwithstanding the large range of responses, a reproducible and consistent decrease in the LPC response of between 54 and 82% was observed when MVC was co-administered with the CDE diet.

As MVC appeared to modulate LPC abundance, we sought to define its effect on proliferation. Phosphorylation of STAT3, ERK and AKT is downstream of CCR5 signalling and intimately involved in cell proliferation and hepatic regeneration [39,40], suggesting that modified pAKT- signalling may play a role in attenuating the LPC response. The levels of pAKT and pSTAT3 trended higher in CDE liver and diminished significantly by MVC administration, indicating CCR5 mediates the LPC response and MVC attenuates it. Surprisingly, changes in pAKT and pSTAT3 were not detected in LPCs or BMMOs, nor did CCL5 treatment alter their proliferation. Collectively these data indicate that MVC modulates active pAKT and pSTAT3 signalling in the CDE diet; however, these proliferative effects are not directly achieved within the LPC or BMMO compartment.

It has been reported that LPCs express CCR5 [21] and they increase the number of HSCs through an epithelial–mesenchymal transition process, thus contributing to liver fibrosis [51]. In contrast to these findings, we were unable to detect CCR5 expression in freshly isolated primary LPC cultures, or LPC cell lines, using RNA-seq and RT-PCR. High levels of CCR5 expression were detected in control liver; however, considerably lower expression was detected in primary hepatocytes. These results indicate that neither hepatocytes nor LPCs are the main contributors of CCR5 expression in the CDE liver. Macrophages are crucial drivers of tumour progression, and they express CCR5 [52], consistent with our finding that BMMOs express high levels of CCR5.

The expression of CCR5 was also studied by IHC in liver sections [28] of CDE and CDE+MVC groups. In animals receiving the CDE diet, CCR5 expression was upregulated around portal tracks, mostly in inflammatory cells such as macrophages and HSCs. Moreover, our snRNA-seq data show that only inflammatory cells display CCR5. The snRNA-Seq data show that cells with the expression profile of LPCs do not express CCR5 mRNA (or levels are not detectable). 

MVC inhibits T-cell proliferation [53]. It also inhibits T-cell chemotaxis toward their cognate ligands [54] and reduces the regulatory T-cell frequency [55], as well as leukocyte trafficking to the gut [56], cancer-associated fibroblasts [57] and macrophage infiltration [58]. It has also been reported that MVC decreases chemotactic activity of macrophages [59,60]. Moreover, MVC can decrease AKT signalling as demonstrated in both RAW 264.7 cells and bone-marrow-derived macrophages [61]. Therefore, MVC might indirectly attenuate the LPC response by affecting other cell types, particularly, macrophages. Consistent with this proposition, MVC-treated BMMOs showed lower migration capabilities in comparison with CCL5-treated cells. This mechanism is a likely explanation for the reduced liver tumour formation/progression elicited by MVC previously reported [28]. 

The inflammatory response is a major component of the CDE dietary model [62]. Our finding that CCR5 is expressed in whole liver and is increased in CDE-treated liver suggests that liver inflammatory cells are the main source of the receptor. HSC activation induced by the CDE diet can be pro-inflammatory, supporting the development of HCC [63]. Our previous study demonstrated that MVC can modulate HSC activity and reduce the levels of proinflammatory cytokines such as IL-6, IL-12, TGF-β and MMP9, suggesting a reduced recruitment of immune cells [28]. In this study, we confirmed the activity of CCR5 on immune cell recruitment by demonstrating that infiltrating CD45+ decreased in number in response to MVC.

Macrophage subpopulations are either classically activated (M1; pro-inflammatory/tumouricidal) or alternatively activated (M2; specialised to suppress inflammation) [64]. Tumour-associated macrophages (TAMs) are a heterogeneous population of myeloid cells that contribute to immunosuppression, favouring the establishment and persistence of solid tumours as well as metastatic dissemination. Like regular macrophages, TAMs can be classified on an M1-to-M2 spectrum, expressing variable levels of cytokines. Tissue microenvironment (TME) ligands, including CCL5, recruit TAMs to the TME [65]. Monocytes from peripheral blood are recruited into the TME and differentiate into TAMs in response to chemokines, including CCL5 (secreted by M2-like macrophages), and growth factors produced by stromal and tumour cells [64]. By secreting CCL5, M2-TAMs may activate signal STAT3 leading to enhanced cancer cell proliferation and invasion/metastasis formation. Upon secreting CCL5, tumour cells recruit CCR5-expressing TAMs [66,67].

Concordantly, Halama et al. [68] found that blocking the CCR5/CCL5 axis with MVC in functional organoids derived from metastatic colorectal cancer (CRC) patients induced macrophage repolarization with anti-tumoural effects. Immunosuppressive M2-tumour-associated macrophages (TAMs) can be reprogrammed to an anti-tumoural M1-TAM subtype by targeting the CCR5/CCL5 axis [69]. Consistently, a significant positive correlation was found between the expression of CCL5 and CD68 in gastric cancer tissues. High levels of CCL5 and CD68 are associated with tumour size, degree of tumour invasion, lymphatic metastasis and pathological grading [70,71].

We demonstrated that MCV reduces liver immunosuppressive M2-TAM markers (CD68 and YM1). Therefore, we suggest that MVC, by reducing the binding of CCL5 to CCR5, is able to inhibit TAM promotion of tumour cell proliferation, invasion and metastasis. Targeting the CCR5/CCL5 axis with MVC reprograms immunosuppressive M2-TAMs to anti-tumoural M1-TAMs, reinvigorating anti-tumour immunity. Thus, MVC, which is capable of disrupting CCL5/CCR5 interaction, may represent a new potential therapeutic option to counteract TAM-induced tumourigenesis. The inhibition of CCL5 secretion by cancer cells or by the TME may represent an additional avenue to counteract liver tumour progression.

## 5. Conclusions

MVC attenuates the development of HCC in the mouse CDE model. It affects both the incidence of HCC and the size of tumours [28]. We show that LPCs themselves do not express CCR5, and thus MVC does not attenuate the LPC response by direct action on these cells. Instead, its action is likely mediated by its ability to temper liver inflammation, which drives the LPC response. The anti-tumorigenic effect of MVC is two-fold. First it attenuates the LPC response and second, it reduces the pro-carcinogenic state of the macrophages by reducing those with an M2 phenotype. As MVC is an approved drug for treating HIV patients, this study suggests that it may be re-purposed for reducing the progression of HCC in chronic inflammatory liver pathologies such as alcoholic liver disease, NAFLD, HBV and HCV, which show high numbers of LPCs, and where the LPC response or signature correlates with disease severity and prognosis [5].

## Figures and Tables

**Figure 1 cancers-13-04935-f001:**
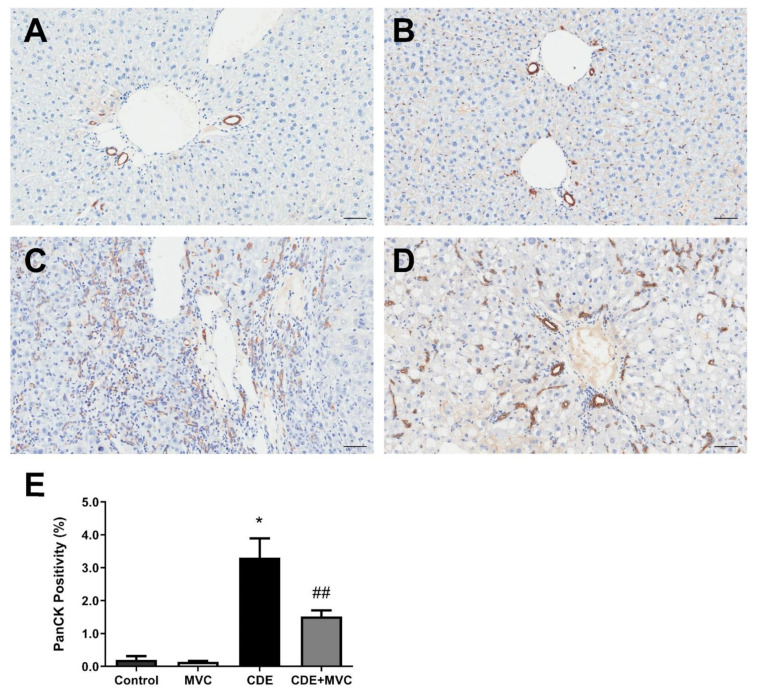
PanCK-positive cell numbers are reduced by MVC in CDE liver. (**A**–**D**) Representative images of histological sections stained with PanCK antibody, from 16-wk control (**A**), MVC (**B**), CDE (**C**) and CDE+MVC (**D**) animal groups. (**E**) Quantitation of PanCK staining by pixel positivity. Bars represent means + SEM for *n* = 4, 8, 4 and 8 in the control, MVC, CDE and CDE+MVC groups, respectively. * *p* < 0.05 compared to control. ## *p* < 0.01 compared to CDE group. Scale bars represent 50 µm.

**Figure 2 cancers-13-04935-f002:**
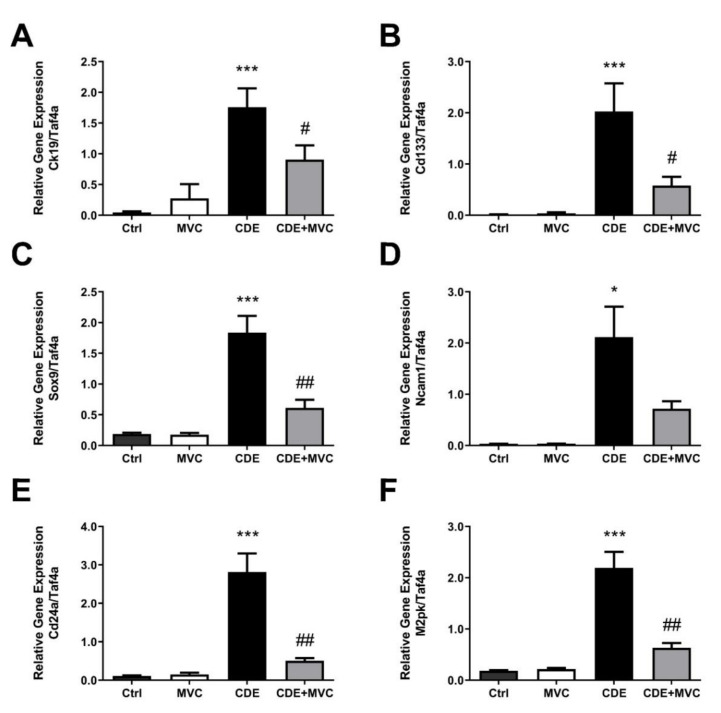
The mRNA expression of LPC markers is attenuated by MVC in CDE liver. (**A**–**F**) Quantitative PCR was performed using cDNA from the livers of animals in the 16-wk control, MVC, CDE, and CDE+MVC groups to amplify genes associated with oval cells. Expression levels are shown relative to the housekeeping gene, TaF4a. Bars represent means + SEM for 7 animals per group. * *p* < 0.05, *** *p* < 0.001 compared to ctrl. # *p* < 0.05, ## *p* < 0.01 compared to CDE group.

**Figure 3 cancers-13-04935-f003:**
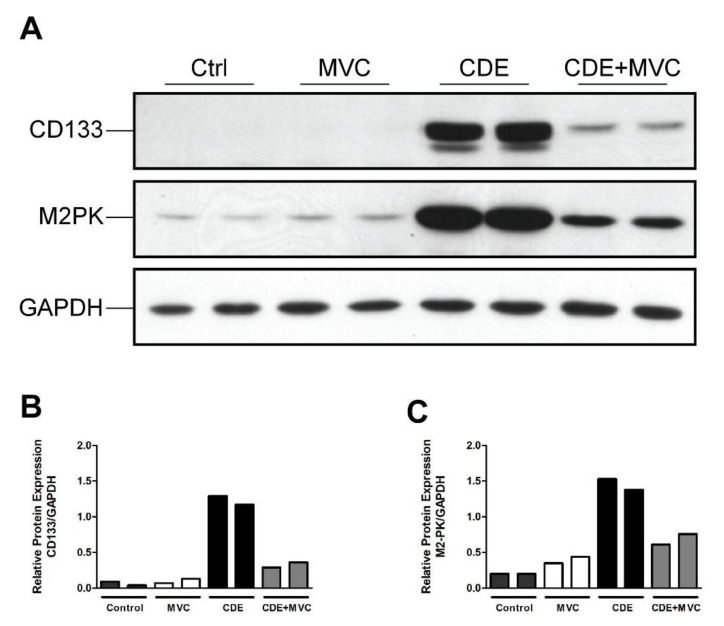
The protein expression of LPC markers is attenuated by MVC in CDE liver. (**A**) Protein lysates from livers of two mice fed either control, CDE, MVC or CDE+MVC diets for 16 weeks were immunoblotted and labelled for CD133, M2PK and the loading control GAPDH. (**B**,**C**) Abundance of CD133 and M2PK relative to GAPDH, as calculated through densitometry performed on the images in (**A**).

**Figure 4 cancers-13-04935-f004:**
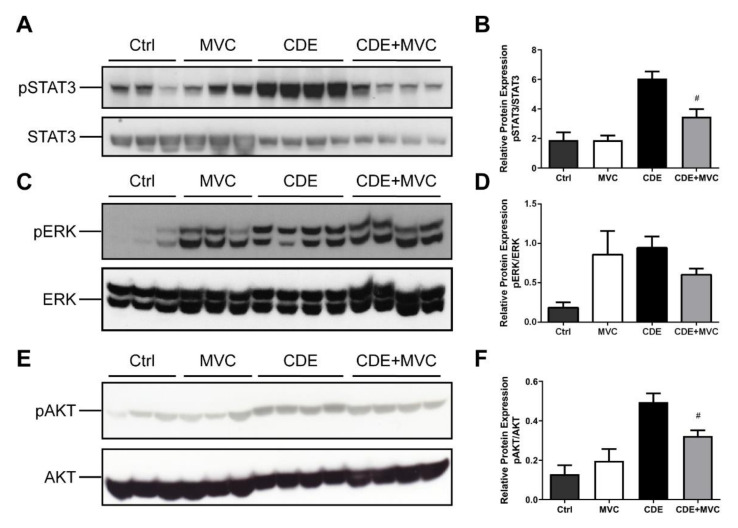
pSTAT3 and pAKT expression is attenuated by MVC in CDE liver. (**A**,**C**,**E**) protein lysates from livers of mice fed Ctrl, CDE, MVC or CDE+MVC diets for 16 weeks were immunoblotted and labelled for phosphorylated (**A**) STAT3, (**C**) pERK, (**E**) pAKT and their loading controls; total STAT3, ERK and AKT, respectively. (**B**,**D**,**F**) the abundance of each phosphorylated protein relative to its control was calculated through densitometry performed on the blot images. Bars represent means + SEM for *n* = 3 mice in the Ctrl and MVC groups and *n* = 4 in the CDE and CDE+MVC groups. # *p* < 0.05 compared to CDE group.

**Figure 5 cancers-13-04935-f005:**
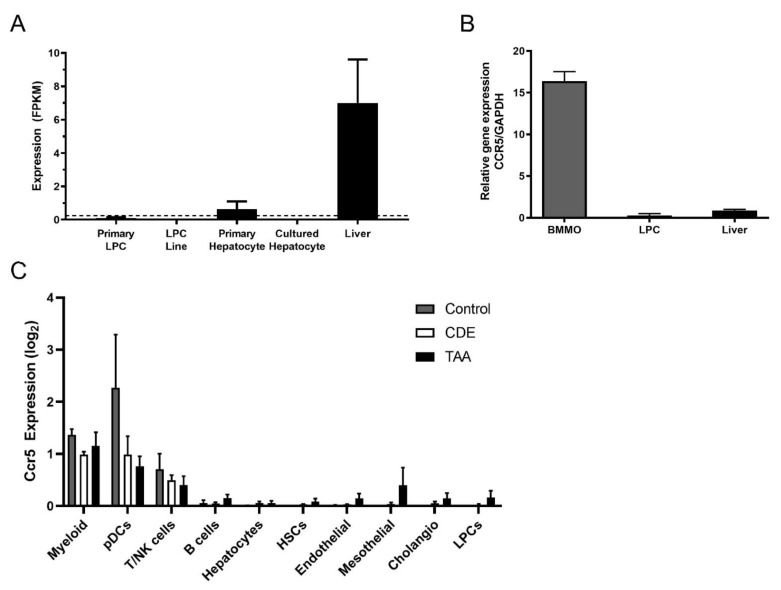
Inflammatory cells express high levels of *Ccr5*, and LPCs do not. (**A**) Expression of *Ccr5* was examined by RNA-Seq in primary LPCs, LPC lines, primary and cultured hepatocytes and whole normal murine liver. Expression values are provided as fragments per kilobase of transcript per million mapped reads (FPKM). A dotted threshold was placed at FPKM = 0.25 to indicate the level above which a gene may be considered actively expressed. Bars represent means + SEM *n* = 3, 12, 3, 3 and 3 for primary LPC cultures, LPC lines, primary hepatocyte cultures, cultured hepatocytes and liver tissue, respectively. (**B**) RT-PCR was performed to amplify *Ccr5* and the GAPDH loading control, using cDNA from BMMOs, LPCs and murine liver. (**C**) Single-cell *Ccr5* expression analysed by snRNA-Seq in Control, CDE-treated and TAA-treated livers. Ten major hepatic cell types were identified and log2-transformed *Ccr5* expression reported for every cell type in each treatment group. Bars represent mean ± SEM. Myeloid (macrophages and monocytes), pDCs (plasmacytoid dendritic cells), T/NK (T cells and natural killer cells), HSCs (hepatic stellate cells), Cholangio (cholangiocytes), LPCs (liver progenitor cells).

**Figure 6 cancers-13-04935-f006:**
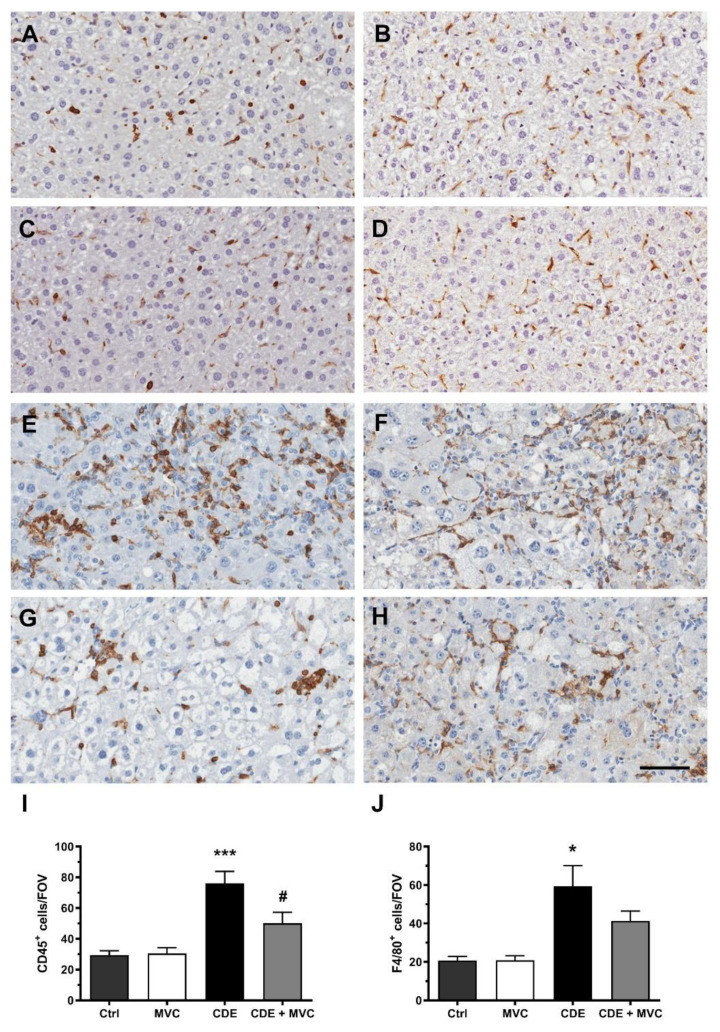
Resident and infiltrating inflammatory cell numbers are attenuated by MVC in CDE liver. Representative images of histological sections stained with CD45 (**A**,**C**,**E**,**G**) or F4/80 (**B**,**D**,**F**,**H**) in Ctrl (**A**,**B**), MVC (**C**,**D**), 16-wk CDE (**E**,**F**) and CDE+MVC (**G**,**H**) animal groups. (**I**,**J**) Quantitation of CD45+ and F4/80+ cell numbers per field of view by the inForm software. Bars represent the mean ± SEM for 8, 11, 6 and 14 animals in the Ctrl, MVC, CDE and CDE+MVC groups, respectively. * *p* < 0.05, *** *p* < 0.001, compared to Ctrl. # *p* < 0.05 compared to CDE group. The scale bar represents 50 µm.

**Figure 7 cancers-13-04935-f007:**
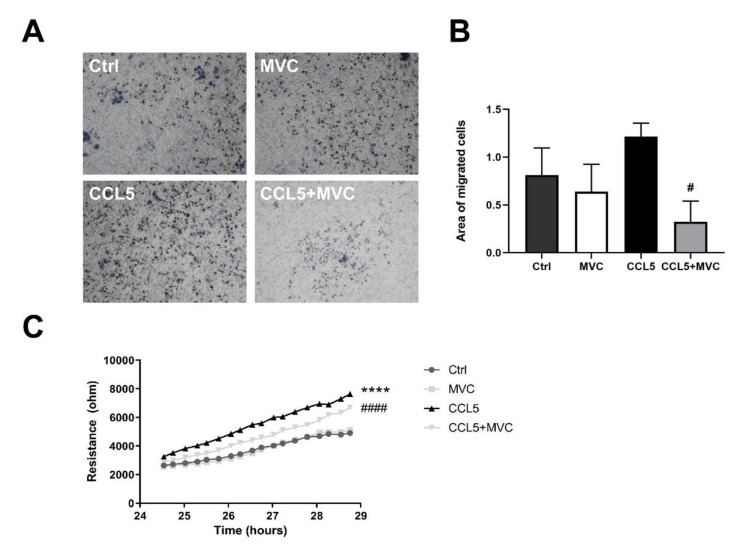
MVC reduces macrophage migration in vitro. (**A**) Transwell migration assay. BMMOs were seeded in the upper chamber of the Transwell ± MVC, and medium without serum but containing mCCL5 was placed in the well. Images display migrated crystal-violet-stained cells following a 24 h migration. Representative images are shown at 10x objective lens magnification. (**B**) Quantitation of the migration images by image thresholding. The bars represent the mean + SEM for *n* = 3, 4, 4 and 4 for the Ctrl, MVC, CCL5 and CCL5+MVC groups, respectively. (**C**) Resistance/impedance over time as measured by the cell migration after electric wounding (ECIS) apparatus. Resistance is proportional to wound closure, representative of cell migration. Cells were stimulated (± MVC and CCL5). Data shown are results from 4 different cell cultures. **** *p* < 0.0001 between Ctrl and CCL5, # *p* < 0.05, #### *p* < 0.0001 between CCL5 and CCL5+MVC.

**Figure 8 cancers-13-04935-f008:**
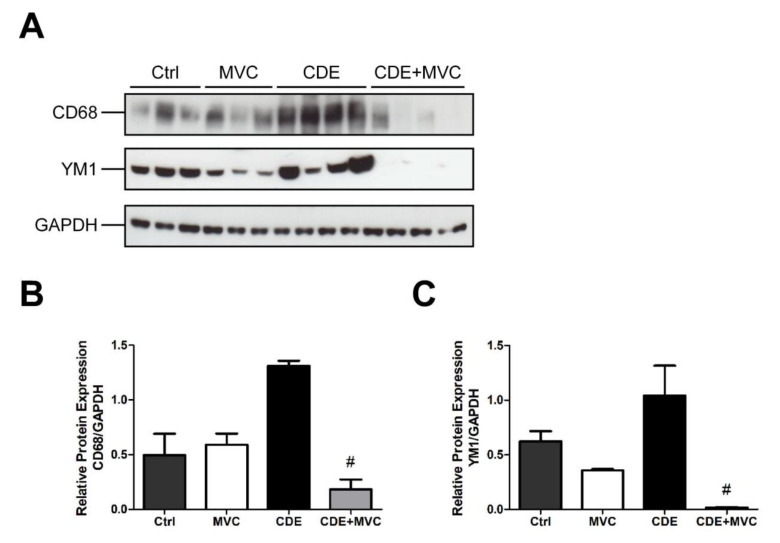
MVC diminishes CDE-induced M2 polarization of macrophages. (**A**) Protein lysates from livers of mice fed Control, CDE, MVC and CDE+MVC diets for 16 weeks were immunoblotted and labelled for CD68, YM1 and the loading control GAPDH. (**B**,**C**) Abundance of each protein relative to its control, was calculated through densitometry performed on the images. Bars represent means + SEM for *n* = 3 mice in the Ctrl and MVC groups and *n* = 4 in the CDE and CDE+MVC groups. # *p* < 0.05 compared to CDE group.

## Data Availability

Data are available upon reasonable request.

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
