# Peer review of "Maraviroc Prevents HCC Development by Suppressing Macrophages and the Liver Progenitor Cell Response in a Murine Chronic Liver Disease Model"

_cancers, 2021, doi:10.3390/cancers13194935_

Round 1
Reviewer 1 Report
Comments
In the article “Maraviroc prevents HCC development by suppressing macro-phages and the liver progenitor cell response in a murine chronic liver disease model,” the authors explore the mechanism by which the drug maraviroc reduces liver progenitor cells (LPCs) in a mouse model of hepatocellular carcinoma. They find that the drug does not directly affect LPCs, but instead alters the phenotype of tumor-associated macrophages, which suppresses inflammation and blocks LPC development.
Overall, this is a well written paper with a sound scientific premise and good experimental evidence. There are, however, a few issues that need to be addressed prior to publication. These are discussed below.
- One of the side effects of maraviroc is liver damage (black box hepatotoxicity warning). This needs to be addressed in the article. Perhaps the authors can speak to the use of maraviroc prophylactically with this in mind, since maraviroc seems particularly ill-suited to the patient population they would seek to treat (individuals with “chronic inflammatory liver pathologies”).
- In section 3.5, it is stated that “LPCs did not display changes in AKT, STAT3 or ERK signalling [sic] in response to CCL5 and/or MVC.” Is this referencing Figure S5? Figure S5 shows a difference in phosphorylated ERK treated with CCL5. Also, the word “signalling” is misspelled [signaling].
- Please include actual p value when reporting statistical significance, not simply p<0.05 (e.g., section 3.2).
- Please be consistent when comparing mean values. In some instances, a percent change is used, in other instances an X-fold change is used (e.g., section 3.3).
- pg 4, paragraph 3, line 3 “in the TNF receptor KO mice” is redundant; delete it.
- pg 5, Figure 5 legend, line 1 and multiple times throughout the legend “Ccr5” should be CCR5. The same issue is on pg. 19 under Supplementary Materials and pg 7 section 2.11.
- Is there a reason that section 3.4 is italicized?
Author Response
We thank the reviewer for their comments. Please find our responses below.
One of the side effects of maraviroc is liver damage (black box hepatotoxicity warning). This needs to be addressed in the article. Perhaps the authors can speak to the use of maraviroc prophylactically with this in mind, since maraviroc seems particularly ill-suited to the patient population they would seek to treat (individuals with “chronic inflammatory liver pathologies”).
No medicine is 100% safe and maraviroc does indeed carry a black box hepatotoxicity warning; however, this may not be scientifically justified. A recent review of the hepatotoxicity of anti-retroviral drugs stated that "the combined clinical trial data and extended evaluation of maraviroc use over five years in close to 1000 patients do not justify the concern prompted by the black box warning" (Otto et al. Hepatotoxicity of Contemporary Antiretroviral Drugs: A Review and Evaluation of Published Clinical Data. Cells. 2021). The article contends that the black box warning applied to maraviroc is in fact due to a blanket CCR5 inhibitor warning applied after earlier hepatotoxic effects in aplaviroc (another CCR5 inhibitor) trials.
In fact, the hepatic safety of MVC has been thoroughly assessed. An investigation was performed across all Pfizer-sponsored maraviroc clinical trials, in which 2350 volunteers received maraviroc. Sporadic hepatic enzyme abnormalities were reported but "demonstrated no dose relationship or association with hyperbilirubinemia". Furthermore, there was no "significant imbalance in hepatic enzyme abnormalities or hepatobiliary adverse events in MVC versus comparator arms up to week 96. The findings were similar in patients coinfected with hepatitis B and/or C virus". In that review, only 2 participants presented a severe hepatotoxicity, although other potential causes were present (Ayoub et al. Hepatic safety and tolerability in the maraviroc clinical development program. AIDS. 2010). In this review, the authors concluded that "MVC does not present significant risks to hepatic safety when taken at the recommended doses in the populations studied". It is important to note that MVC is never employed alone. The hepatic risk could be attributed to other HIV therapies or to another factor such as co-infections (HBV, HCV) or a low CD4-T cell count, indeed MVC still remains as a therapeutic option in HIV-infected persons.
Many other studies have also confirmed the hepatic safety of MVC (Rockstroh et al. Hepatic safety of maraviroc in patients with HIV-1 and hepatitis C and/or B virus: 144-week results from a randomized, placebo-controlled trial. Antivir Ther. 2017; Crespo et al. Hepatic safety of maraviroc in HIV-1-infected patients with hepatitis C and/or B co-infection. The Maraviroc Cohort Spanish Group. Enferm Infecc Microbiol Clin. 2017; Rockstroh Hepatic safety in subjects with HIV-1 and hepatitis C and/or B virus: a randomized, double-blind study of maraviroc versus placebo in combination with antiretroviral agents. HIV Clin Trials. 2015; Gonzalez et al. The effects of Maraviroc on liver fibrosis in HIV/HCV co-infected patients. J Int AIDS Soc. 2014).
Additionally, it is known that CCR5 has pro-fibrogenic effects in hepatic cells. As such, blocking CCR5 co-receptors has been proposed as a therapeutic option (i.e. reducing the release of pro-inflammatory cytokines implicated in liver fibrosis) to avoid the progression of hepatic fibrosis (i.e. in HIV/HCV co-infected patients). Different authors have observed that MVC has a positive effect on liver fibrosis (Gonzalez et al. The effects of Maraviroc on liver fibrosis in HIV/HCV co-infected patients. J Int AIDS Soc. 2014; Coppola et al. Effects of treatment with Maraviroc a CCR5 inhibitor on a human hepatic stellate cell line. J Cell Physiol. 2018).
Finally, in our hands, MVC does not increase transaminases levels (Fig. 1C Ochoa-Callejero et al. Maraviroc, a CCR5 Antagonist, Prevents Development of Hepatocellular Carcinoma in a Mouse Model. PLoS ONE 2013). Furthermore, in the presence of liver damage, maraviroc normalises inflammation levels as determined by cytokine expression by ELISA and qRTPCR-cDNA (Ochoa-Callejero et al. Maraviroc, a CCR5 Antagonist, Prevents Development of Hepatocellular Carcinoma in a Mouse Model. PLoS ONE 2013).
Given the above, our view is that a prophylactic application of maraviroc may be well tolerated and beneficial if taken in the appropriate circumstances.
In section 3.5, it is stated that “LPCs did not display changes in AKT, STAT3 or ERK signalling [sic] in response to CCL5 and/or MVC.” Is this referencing Figure S5? Figure S5 shows a difference in phosphorylated ERK treated with CCL5. Also, the word “signalling” is misspelled [signaling].
Whilst the reviewer is correct in that pERK does indeed significantly increase in response to CCL5, and MVC does not abrogate this response. This lack of response to MVC is what we were expecting to show and we have thus altered the language in section 3.3 and 3.5 to clarify this. Regarding spelling, we had intended to Americanize the spelling in our manuscript and thus convert to use of "signaling" with one 'l'. The reviewer has brought to our attention that we had missed some words and as such, we will instead use "U.K." English throughout the text.
Please include actual p value when reporting statistical significance, not simply p<0.05 (e.g., section 3.2).
We have now made this change throughout the manuscript.
Please be consistent when comparing mean values. In some instances, a percent change is used, in other instances an X-fold change is used (e.g., section 3.3).
We have now converted fold-changes to percentages when comparisons are made.
pg 4, paragraph 3, line 3 “in the TNF receptor KO mice” is redundant; delete it.
The requested change has been made.
pg 5, Figure 5 legend, line 1 and multiple times throughout the legend “Ccr5” should be CCR5. The same issue is on pg. 19 under Supplementary Materials and pg 7 section 2.11.
We have endeavoured to use the correct nomenclature throughout the manuscript. In this case, where we are references mouse genes, the correct nomenclature is to in fact capitalise the first letter and use lowercase for the remaining letters. It seems we have however neglected to italicise these mouse genes and have now made this change throughout the manuscript.
Is there a reason that section 3.4 is italicized?
It's likely that the paragraph got caught in the formatting of the section title when formatting within the MDPI template. This has now been fixed.
Reviewer 2 Report
The authors demonstrated that Maraviroc, a drug approved for treating HIV patients, reduces the liver stem cell response and macrophage activation in a liver cancer mouse model using various methods. These findings underline the preventive potential of this drug in liver cancer.
This study is well designed, conducted and presented.
Some minor points in Materials and Methods:
Section 2.1 Animal model line 146 C57/BL6 mouse, Need a bit more information of this genotype of mouse.
Section 2.7 RNA extraction and gene expression quantification, need to specify which genes are tested and rationale to choose these genes.
Section 2.8 Western blotting and protein quantitation, need to specify which proteins are tested and rationale to choose these proteins.
Section 2.9 Proliferation assay, need to explain more on how proliferation assay is calculated.
Section 2.13 Transwell migration assay Line 288, typo "5 X 104" should be 5 x 10 power of 4.
Section 2.14 β-. arrestin recruitment. Typo? should be "β-arrestin".
Results: Figure 5C is too small.
Author Response
We thank the reviewer for their comments. Please find our responses below.
Section 2.1 Animal model line 146 C57/BL6 mouse, Need a bit more information of this genotype of mouse.
We have now referred to the mice as WT C57BL/6J mice.
Section 2.7 RNA extraction and gene expression quantification, need to specify which genes are tested and rationale to choose these genes.
In section 2.7, we reference the table containing the primer sets used for gene expression. Part of the rationale is built into the introduction but more explicitly, mention the specific genes and their rationale for use can already be found in results section 3.2, pg 9. We believe the results sections is the most appropriate location and do not believe this should be replicated in the methodology.
Section 2.8 Western blotting and protein quantitation, need to specify which proteins are tested and rationale to choose these proteins.
In section 2.8, every antibody used for Western blotting is already specified when we outline the concentrations used. As with our response to the comment above, some soft rationale is provided in the introduction but full rationale for use of these antibodies is given within the relevant results sections. These can be seen in sections 3.2, 3.3 and 3.8 pages 10, 11 and 12 respectively.
Section 2.9 Proliferation assay, need to explain more on how proliferation assay is calculated.
We have updated this section so it is clear.
Section 2.13 Transwell migration assay Line 288, typo "5 X 104" should be 5 x 10 power of 4.
We thank the reviewer for pointing out this error and have now searched elsewhere in the manuscript to make the correction globally. We have now amended the manuscript to correct these errors.
Section 2.14 β-. arrestin recruitment. Typo? should be "β-arrestin".
We have now amended the manuscript to correct this error.
Results: Figure 5C is too small.
We thank the reviewer for this critique and have now increased the size of this panel by 25%.